# Abnormal Functional Network Topology and Its Dynamics during Sustained Attention Processing Significantly Implicate Post-TBI Attention Deficits in Children

**DOI:** 10.3390/brainsci11101348

**Published:** 2021-10-13

**Authors:** Meng Cao, Jeffery M. Halperin, Xiaobo Li

**Affiliations:** 1Department of Biomedical Engineering, New Jersey Institute of Technology, Newark, NJ 07102, USA; meng.cao@njit.edu; 2Department of Psychology, Queens College, City University of New York, New York, NY 11367, USA; jeffrey.halperin@gmail.com

**Keywords:** pediatric, traumatic brain injury (TBI), attention deficits, functional MRI (fMRI), graph theory, dynamic functional connectivity

## Abstract

Traumatic brain injury (TBI) is highly prevalent in children. Attention deficits are among the most common and persistent post-TBI cognitive and behavioral sequalae that can contribute to adverse outcomes. This study investigated the topological properties of the functional brain network for sustained attention processing and their dynamics in 42 children with severe post-TBI attention deficits (TBI-A) and 47 matched healthy controls. Functional MRI data during a block-designed sustained attention task was collected for each subject, with each full task block further divided into the pre-, early, late-, and post-stimulation stages. The task-related functional brain network was constructed using the graph theoretic technique. Then, the sliding-window-based method was utilized to assess the dynamics of the topological properties in each stimulation stage. Relative to the controls, the TBI-A group had significantly reduced nodal efficiency and/or degree of left postcentral, inferior parietal, inferior temporal, and fusiform gyri and their decreased stability during the early and late-stimulation stages. The left postcentral inferior parietal network anomalies were found to be significantly associated with elevated inattentive symptoms in children with TBI-A. These results suggest that abnormal functional network characteristics and their dynamics associated with the left parietal lobe may significantly link to the onset of the severe post-TBI attention deficits in children.

## 1. Introduction

Pediatric traumatic brain injury (TBI) is a significant public health issue, which occurs in more than 3 million children each year globally [1]. Neurocognitive impairments and behavioral abnormalities, including attention problems, depression and mood disorders, anxiety, and posttraumatic stress disorder, were frequently reported in children with chronic TBI [2,3,4,5]. Attention deficits are among the most common and persistent cognitive and behavioral consequences that can be observed in at least 35% of children within two years of their injuries [6,7]. Childhood post-TBI attention deficits (TBI-A) have been found to link with a significantly heightened risk for the development of severe psychopathology and impairments in overall functioning in late adolescence [8,9]. The neural substrates associated with TBI-A in children have not yet been well investigated. Understanding the early brain mechanisms of TBI-A have considerable heuristic value for informing novel and timely strategies of prevention and intervention in affected individuals.

The blood-oxygen level-dependent (BOLD) response-based functional MRI (fMRI) has been widely implemented to examine the neurophysiological alterations associated with TBI-related functional brain damages [10]. The majority of task-based fMRI studies in TBI have used working memory paradigms and have reported abnormal functional activation in cortical and subcortical areas, such as the prefrontal cortex, the superior temporal gyrus, and hippocampus, which were associated with working memory impairments in subjects with TBI [11,12,13,14], while a recent study reported no significant working memory-related brain differences between children with TBI and controls [15]. Other fMRI studies have also reported functional alterations in frontal, parietal, temporal, and occipital regions in children with TBI during motor tasks [16], language tasks [17], and social cognition tasks [18]. Only a few pediatric TBI studies have investigated brain activations during attention-related tasks [19,20,21,22]. Studies have reported that during sustained attention processing, children with TBI demonstrated reduced activations in frontal, parietal, and occipital regions when compared with healthy controls [22] and children with orthopedic injuries [19]. Tlustos and colleagues reported that children with TBI, relative to controls, showed decreased activation in anterior cingulate and motor cortex during inhibitory control processing [20]. Children with TBI also showed hyperactivations in the middle frontal gyrus, the precentral gyrus, and the parietal lobule during interference control processing [21]. The discrepancies of findings from the existing studies might be partially explained by the differences of task design, techniques implemented for data analyses, sample size, environmental factors, and subject-related biases without controlling the heterogeneity of neurocognitive/behavioral outcomes induced by TBI [23].

Resting-state fMRI (rs-fMRI) studies in TBI have also reported inconsistent results. Relative to matched controls, children with chronic TBI have been found to have reduced functional connectivity (FC) between caudate and motor network [24], increased FC between the frontal and fusiform gyrus [25], or reduced FC between rostral anterior cingulate cortex and amygdala [26]. In addition, graph theoretical technique (GTT)-based rs-fMRI studies have reported system-level topological alterations in adults with TBI, relative to controls [27]. For instance, studies have shown decreased functional network integration, including decreased network global efficiency and increased characteristic path length, in adult with TBI when compared with healthy controls [28,29,30]. A more recent longitudinal study reported that increased functional integrations were associated with better overall cognitive recovery in adults with TBI [31]. With the help of the advances in recent techniques and methodologies, dynamic FC patterns during both resting-state and cognitive processes have been increasingly observed and linked to neurobehavioral variations in normal controls and subjects with mental disorders [32,33,34,35]. Relative to the procedure of constructing the functional network using overall time duration (referred to as static functional network), dynamic analysis evaluates the functional network topology at a temporal basis. Gilbert et al. reported that during a working memory and information processing task, adults with TBI demonstrated more brain states than controls but with less between-state transitions [35]. An rs-fMRI study found both static and dynamic alterations in adult with TBI at the acute stage, which were associated with persistent symptoms at a chronic stage [36].

Indeed, the GTT- and dynamic FC-based investigations in functional brain networks during resting state and cognitive tasks have allowed a new dimension in understanding the neural mechanisms associated with post-TBI neurocognitive and behavioral impairments in adults. However, the system-level functional brain organizations, their temporal dynamics, and their associations with TBI-related cognitive/behavioral deficits have not yet been sufficiently revealed in children. The current study proposed to utilize the GTT- and dynamic FC-based techniques to study the topological properties and their dynamics of the functional network for attention processing, and their relations with TBI-related attention deficits in a homogeneous group of children with TBI-A and matched controls. Our previous research has showed disrupted structural network topological properties in frontal, parietal, and temporal regions in children with TBI-A [37]. Based on the existing findings from our and other groups, we hypothesize that relative to matched controls, children with TBI-A may exhibit significantly altered topological properties and their dynamic features in frontal, parietal, and temporal areas, and these system-level anomalies in the functional network for attention processing strongly link to the severe attention problems in children with TBI-A.

## 2. Materials and Methods

### 2.1. Participants

A total of 89 children, including 42 children with TBI-A and 47 controls were involved in the study. The TBI-A subjects were recruited from the New Jersey Pediatric Neuroscience Institute, Saint Peter’s University Hospital, and local communities in New Jersey. Controls were solicited from the local communities by advertisement in public places. The study received institutional review board approval at the New Jersey Institute of Technology and Saint Peter’s University Hospital. Prior the study, all the participants and their parents or guardians provided written informed assent and consent, respectively.

The inclusion criteria for the TBI-A group were: (1) history of clinically diagnosed mild-to-moderate non-penetrating TBI with the severity scores ranging from 9 to 15 using the Glasgow Coma Scale (GCS) [38] and no overt focal brain damages or hemorrhages; (2) the first TBI incidence happened at least 6 month prior to the study; (3) T score ≥ 65 in inattention subscale, hyperactivity subscale, or both in the Conners 3rd Edition—Parent Short form (Conners 3–PS) [39] assessed during the study visit. In addition, subjects with a history of diagnosed attention-deficit/hyperactivity disorder (ADHD) (any sub-presentations) prior the diagnosis of TBI, or severe pre-TBI inattentive and/or hyperactive behaviors that were reported by a parent were not included, to minimize con-founding factors. The control group included children with (1) no history of TBI; (2) no history of diagnosed ADHD (any sub-presentation); (3) T-scores ≤ 60 in all the subscales in the Conners 3–PS assessed during the study visit. The two groups were matched on age, sex (male/female) distribution, and socioeconomic status (SES) (estimated using the average education year of both parents).

To further improve the homogeneity of the study sample, the general inclusion criteria for both groups included (1) only right-handed, to remove handedness-related potential effects on brain structures; (2) full scale IQ ≥ 80, to minimize neurobiological heterogeneities in the study sample; (3) ages of 11–15 years, to reduce neurodevelopment-introduced variations in brain structures. In the current study, handedness was evaluated using the Edinburgh Handedness Inventory [40]. Full scale IQ was estimated by the Wechsler Abbreviated Scale of Intelligence II (WASI-II) [41]. None of the subjects involved in this study had (1) current or previous diagnosis of Autism spectrum disorders, pervasive development disorder, psychosis, major mood disorders (except dysthymia not under treatment), post-traumatic stress disorder, obsessive compulsive disorder, conduct disorder, anxiety (except simple phobias), or substance use disorders, based on Diagnostic and Statistical Manual of Mental Disorders 5 (DSM-5) [42] and supplemented by the Kiddie Schedule for Affective Disorders and Schizophrenia for School-Age Children-Present and Lifetime Version (K-SADS-PL) [43]; (2) any types of diagnosed chronic medical illnesses, neurological disorders, or learning disabilities, from the medical history; (3) treatment with long-acting stimulants or non-stimulant psycho-tropic medications within the past month; (4) any contraindications for MRI scanning, such as claustrophobia, tooth braces or other metal implants. In addition, pre-puberty subjects were also excluded to reduce confounders associated with different pubertal stages [44]. Puberty status was evaluated using the parent version of Carskadon and Acebo’s self-administered rating scale [45]. After initial processing of the neuroimaging data from each subject, 3 subjects were excluded from further analyses due to heavy head motion. Therefore, a total of 40 patients with TBI-A and 46 controls were included in group-level analyses. All the demographic and clinical measures were summarized in Table 1.

### 2.2. Neuroimaging Data Acquisition Protocol

MRI scans for each subject were performed on a 3-Tesla Siemens TRIO (Siemens Medical Systems, Erlangen, Germany) scanner at Rutgers University Brain Imaging Center. The fMRI data were acquired using a whole brain gradient echo-planar sequence (voxel size = 1.5 mm × 1.5 mm × 2.0 mm, repetition time (TR) = 1000 ms, echo time (TE) = 28.8 ms, and field of view = 208 mm, slice thickness = 2.0 mm). For data co-registration, a high-resolution T1-weighted structural image was also collected with a sagittal multi-echo magnetization-prepared rapid acquisition gradient echo (MPRAGE) sequence (voxel size = 1 mm^3^ isotropic, TR = 1900 ms, TE = 2.52 ms, flip angle = 9°, FOV = 250 mm × 250 mm, and 176 sagittal slices).

### 2.3. Visual Sustained Attention Task for fMRI

Clinical studies have suggested that sustained attention in children is vulnerable to TBI-induced damages [46,47]. The continuous performance task (CPT) is one of the most widely used tasks to measure sustained attention and was shown to be a robust instrument to challenge the sustained attention in children with TBI [48]. In the current study, all subjects were asked to perform an enhanced CPT, the visual sustained attention task (VAST), during fMRI data acquisition. The VAST is a block-designed task that was established and validated in our previous functional imaging studies for achieving optimal power in maintaining sustained attention and assessing related functional brain pathways in children [49,50,51]. The task contains 5 task stimulation blocks that interleaved with 5 resting blocks, as shown in Figure 1A. Each block lasts for 30 s, with a total scan time of 5 min. Within each task stimulation block, a sequence of 3 single-digit numbers was first shown in red to serve as the target, followed by 9 stimulus sequences in black, when subjects were asked to response if each sequence matches the target. Subjects were instructed to stay focused and respond only after the third number of each sequence was shown. To ensure full understanding of the instructions, practical trials of the task were provided to each subject before the scan session.

### 2.4. Individual-Level Neuroimaging Data Pre-Processing

Pre-processing of each set of fMRI data was carried out using the FEAT Toolbox from FMRIB Software Library v6.0 (FSL) [52]. The data were first manually checked for any missing volumes and heavy head motions. Then, the motion artifacts were corrected using rigid-body transformation by registering all volumes to the first volume. The motions for each subject were measured by extracting the six translational and rotational displacement parameters. Due to the critical impacts of the head motions on the construction of both static and dynamic functional networks [53,54], we applied a strict cutoff threshold of 1.5 mm. Three subjects (2 TBI-A subjects and 1 control) were excluded due to heavy head motion. After corrected for slice timing, fMRI data of each subject were then smoothed with a 5 mm full-width at half maximum gaussian kernel to improve the signal-to-noise ratio. A high-pass filter was applied to the time series to remove the low frequency noise and signal drifting. Finally, the fMRI data were co-registered to a MNI152 template, with a voxel size of 2 mm × 2 mm × 2 mm, using each subject’s T1-weighted structural image. Hemodynamic response to task-related condition was modeled using the general linear model, with 24 motion parameters, including the 6 basic displacement parameters (R_t_), and the derivatives (R_t_′) and squares (R_t_^2^ and R_t−1_^2^, where t and t −1 refer to current and preceding timepoints) of these parameters, and nuisance signals (white matter, cortical spinal fluid, and global signal), as additional regressors. The Z statistic images were thresholded using clusters determined by Z > 2.3 and a cluster-based method for multiple comparison correction at *p* < 0.05 [52].

### 2.5. Network Node Selection and BOLD Signal Extraction

To select the nodes for functional network construction, a combined activation map of two groups were first generated and parcellated into 118 cortical and subcortical regions using automatic anatomical labeling (AAL) atlas [55]. A network node was defined as a spherical region of interest (ROI) with the radius of 5 mm in a parcellated region if it has a cluster of at least 100 significantly activated voxels, and centered at the highest local maximum in the cluster. At the end, a total of 59 network nodes were generated (Figure 1B).

The BOLD time series of each node was extracted from the preprocessed fMRI data by averaging the BOLD responses of the voxels in the node. The averaged signal was then decomposed into 5 levels using maximal overlap discrete wavelet transform [56]. Wavelet levels 3, 4, and 5, corresponding to frequency band of 0.015–0.124 Hz, were used to reconstruct the filtered signal to further minimize motion artifacts and non-relevant signals. This selected frequency band had been demonstrated to contain most task-related hemodynamic information [49,50,57].

### 2.6. Static Functional Network Construction

To construct the overall functional brain network responding to the sustained attention processing task, pairwise Pearson’s correlation coefficients of the BOLD signals in the 59 network nodes were first calculated to form the 59 × 59 FC matrix. The matrix was then binarized by thresholding using the network cost, which was defined as the fraction of existing edges relative to all possible edges within a network. To determine the proper threshold range for functional network construction, the network global efficiency and network local efficiency were calculated over the cost range from 0.1 to 0.5, with a step size of 0.01. The network global efficiency is a metric of the network integration that reflects the ability of information transferring across distributed brain areas [58]. It was the average of the inversed shortest distance between each node pair in the network, which is defined as:(1)$$E_{glob}\left(G\right)=\frac{1}{n\left(n-1\right)}\sum_{i,j\in G,j\neq i}\frac{1}{d_{ij}},$$
where n is the number of nodes in the network, and $d_{ij}$ is the inverse of the shortest path length (number of edges) between node i and j. The network local efficiency estimates the network segregation and represents the fault tolerance level of the network [58]. The network local efficiency is the average nodal local efficiency of all nodes in the network, where the nodal local efficiency was defined as the network global efficiency of the subnetwork that consisted of all neighbor nodes of that specific node. This can be calculated using the following formula:(2)$$E_{network-loc}\left(G\right)=\frac{1}{n}\sum_{i\in G}E_{glob}\left(G_{i}\right),$$
where $G_{i}$ is the subnetwork that consists of all neighbor nodes of node i, and the global efficiency of subnetwork $G_{i}$ is calculated using Equation (1). Then, both global metrics of the constructed network were compared with the node- and degree-matched regular and random networks. The functional brain networks in human brain have been proved to be small-world networks that provide high global and local efficiency of parallel information processing while maintaining lower network cost [59]. A network is considered to be small-world if it meets the following criteria: $E_{glob}\left(G_{regular}\right)<E_{glob}\left(G\right)<E_{glob}\left(G_{random}\right)$ and $E_{network-loc}\left(G_{random}\right)<E_{network-loc}\left(G\right)<E_{network-loc}\left(G_{regular}\right)$, where $E_{glob}\left(G_{regular}\right)$, $E_{glob}\left(G_{random}\right)$, $E_{network-loc}\left(G_{regular}\right)$, and $E_{network-loc}\left(G_{random}\right)$ represent the network global efficiency and network local efficiency of the node- and degree-matched regular and random networks, respectively [59]. The proper cost range for functional network construction in both TBI-A and control groups was from 0.15 to 0.45, as shown in Figure 1C. The proportional thresholding method sets arbitrary thresholds in continuous data, which artificially inflate differences in network topology and can arise differences in overall functional connectivity in case–control studies [60]. To validate the eligibility of the proportional thresholding method in our study, the group differences of network global efficiency, network local efficiency, and overall functional connectivity were compared at each network cost step. No significant between-group differences were found in these measures within the selected cost range.

The global and regional topological properties of the overall functional brain network from each subject were then calculated and averaged over the cost range, including the network/nodal global efficiency, network/nodal local efficiency, network/nodal clustering coefficient, nodal degree, and betweenness centrality.

The nodal global efficiency of a specific node is a measure of its nodal communication capacity with all other nodes in the network. It was defined as:(3)$$E_{nodal}\left(i\right)=\frac{1}{n-1}\sum_{j\in N,j\neq i}\frac{1}{d_{ij}},$$
where N contains all the neighbors of node i.

The nodal clustering coefficient describes the likelihood of whether the neighboring nodes of a node are interconnected with each other [61], which was defined as:(4)$$C\left(G\right)=\frac{1}{n}\sum_{i\in G}\frac{1}{k_{i}\left(k_{i}-1\right)}\times \sum_{j,h\in G_{i}}{\left(a_{ij}a_{ih}a_{jh}\right)}^{1/3},$$
where $a_{ij}$ is the connection between node i and j (1 for connected and 0 for not connected), and $k_{i}$ is the number of neighbors of node i.

The betweenness centrality measures the ability for one node to bridge indirectly connected nodes by counting the number of shortest paths that pass through a certain node [62]. The betweenness centrality was defined as: (5)$$B\left(i\right)=\frac{1}{\left(n-1\right)\left(n-2\right)}\sum_{j,k\in N,\;j\neq k}\frac{p(i\;|\;j,k)}{P\left(j,k\right)},$$
where j,k are node pairs in the network. p(i | j,k) is whether the shortest path between node j and node k passes through node i. P(j,k) is the total number of unique shortest path between node j and node k. For each node in a functional brain network, its nodal global efficiency represents the integration of its associated subnetworks, whereas its nodal local efficiency and nodal clustering coefficient represent the modularity [63,64]. All network topological property calculation were performed using the Brain Connectivity Toolbox [65].

### 2.7. Analysis of Functional Network Dynamics

A sliding-window approach was used to investigate the functional network dynamics during the task procedure. A temporal window was defined to include 17 consecutive volumes in the fMRI data. Therefore, a total of 284 temporal windows were generated, with a sliding-step of 1 TR applied along the 300 TRs during the entire task period. For each of the 284 temporal windows, a 59 × 59 FC matrix was formed by the pairwise Pearson’s correlation coefficients of the 59 network nodes. Each contained only 17 time points within that temporal window.

Based on the task design, the task duration consisted of pre-, early, late-, and post-stimulation stages (Figure 1A). The pre-stimulation stage was defined as the 15 s (15 TRs) right before each task-stimulation block. The early stimulation stage was defined as the first 15 TRs of each task-stimulation block, and the late-stimulation stage was defined as the last 15 TRs of each task-stimulation block. The post-stimulation stage was defined as the 15 TRs after each task-stimulation block. Based on the estimated hemodynamic response function, the early stimulation stage corresponds to the recruiting response stage while the late-stimulation stage corresponds to the stable response stage, also shown in Figure 1A. During the network dynamics analyses introduced in the following paragraphs, the temporal window-based FC matrices of the early and late-stimulation stages were extracted from only the first four task-stimulation blocks, to match the involved duration of the pre- and post-stimulation stages. 

A temporal window-based functional network for each sliding step was constructed using the same strategy for static network construction introduced in Section 2.6. The thresholding for network topological property estimations in this step utilized Pearson’s correlation coefficients ranging from 0.55 to 0.85, which corresponded to the top 45% to the top 15% strongest FCs among the 284 FC matrices. Selection of such threshold range can preserve the temporal fluctuation of the overall functional connectivity, while ensuring that the overall edge density of the current network matches that in the static functional network. The network topological properties, including global efficiency, local efficiency, and clustering coefficient at both network and nodal levels, plus degree and betweenness centrality at nodal level, were then calculated for the functional network in each temporal window. The mean and the standard deviation values of each network property were calculated among the temporal networks involved in each of the four task-stimulation stages. The standard deviation characterizes the stability of the network property, which was defined as:(6)$$S=\sqrt{\frac{1}{N-1}\sum_{i=1}^{N}{\left|A_{i}-\mu \right|}^{2}},$$
where *N* is the number of steps in a stage, $A_{i}$ is the specific topological property at step i, and μ is the mean of the topological property over all steps in a stage.

### 2.8. Group-Level Analyses

Group statistics were carried out using R 4.0.3 on macOS Mojave 10.14.1. Between-group comparisons in demographic, clinical, behavioral, and neurocognitive performance measures were conducted using a chi-squared test for categorical data (sex and ethics), and an independent two-sample *t*-test for numerical measures.

Group comparisons in the static network topological properties were performed using a mixed-effects general linear model by setting TBI-A and controls as group variables, and adding IQ, age, SES as random-effect, and sex as fixed-effect covariates, respectively. Topological properties that showed significant between-group differences (corrected using false discovery rate) were selected for post hoc independent-sample *t*-test to provide directional comparisons. Group comparisons in the static network topological properties were controlled for potential multiple comparisons, using the Bonferroni correction with a threshold of significance at corrected α ≤ 0.05 [66].

To test the group difference in sub-stages and the transition between each adjacent stage-pair, a mixed-model analysis of covariance (ANCOVA) was conducted with topological properties at adjacent stages as repeated measure, sex as fixed-effect covariate, and IQ, age, SES as random-effect covariates. Topological properties that showed significant group effects or significant group-stage interaction (corrected using false discovery rate) were selected for post hoc analysis. Group comparison of the selected measures were performed using independent-sample *t*-tests as the post hoc analysis. Post hoc analysis was controlled for potential multiple comparisons (in the total of 4 stages), using the Bonferroni correction with a threshold of significance at corrected α ≤ 0.05 [66]. 

Brain–behavior associations in the TBI-A group were assessed using Pearson’s correlation between the T-scores of the inattentive and hyperactive/impulsive subscales from Conners 3–PS and the network measures that showed significant between-group differences. The correlation analyses were controlled for potential multiple comparisons (in the total number of comparisons), by using the Bonferroni correction with a threshold of significance at corrected α ≤ 0.05.

## 3. Results

### 3.1. Demographic and Clinical/Behavioral Measures

No demographic information was found to have significant group difference. In addition, children with TBI-A did not demonstrate any significant differences in the performance measures of VSAT when compared with controls. Relative to the controls, the children with TBI-A showed significantly more inattentive (t = −16.366, *p* < 0.001) and hyperactive/impulsive (t = −6.835, *p* < 0.001) symptoms measured using the T scores in Conners 3–PS. The demographic information was shown in Table 1.

### 3.2. Topological Measures in Overall Functional Network

Compared with controls, children with TBI-A showed significantly decreased nodal clustering coefficient in the left precentral gyrus (t = 2.653, *p*-_Bonferroni_ = 0.045), and significantly decreased nodal local efficiency (t = 3.770, *p*-_Bonferroni_ = 0.001) and nodal clustering coefficient (t = 3.380, *p*-_Bonferroni_ = 0.005) in the left postcentral gyrus. No significant between-group differences were observed in topological measures at a global level.

### 3.3. Dynamics of the Topological Measures

The network topological properties at each stage were calculated using the average network properties of all steps within each stage. Relative to controls, the TBI-A group showed significantly decreased nodal local efficiency (t = 2.560, *p*-_Bonferroni_ = 0.049) in the left postcentral gyrus at early stimulation stage (Figure 2A); significantly decreased nodal local efficiency (t = 2.798, *p*-_Bonferroni_ = 0.026) and nodal degree (t = 2.603, *p*-_Bonferroni_ = 0.044) in left inferior parietal lobule at early stimulation stage (Figure 2B); significantly decreased nodal local efficiency (t = 2.870, *p*-_Bonferroni_ = 0.021) and nodal clustering coefficient (t = 2.750, *p*-_Bonferroni_ = 0.029) in the left inferior temporal gyrus at late-stimulation stage (Figure 2C); and significantly increased nodal global efficiency (t = −2.702, *p*-_Bonferroni_ = 0.033) in right putamen at late-stimulation stage (Figure 2D).

The stability of each network topological property at each stage was represented by the standard deviation within each stage. Relative to controls, children with TBI-A demonstrated significantly increased standard deviation of network global efficiency at the late-stimulation stage (t = 2.519, *p*-_Bonferroni_ = 0.048) (Figure 3A). At the late stimulation stage, the TBI-A group also showed significantly decreased standard deviations of the right insula nodal local efficiency (t = 3.206, *p*-_Bonferroni_ = 0.007) and nodal clustering coefficient (t = 3.052, *p*-_Bonferroni_ = 0.012) (Figure 3B); significantly increased standard deviations of the left fusiform gyrus nodal local efficiency (t = −2.707, *p*-_Bonferroni_ = 0.033) and nodal clustering coefficient (t = −2.732, *p*-_Bonferroni_ = 0.031) (Figure 3C); and significantly increased standard deviation of the left inferior temporal gyrus nodal local efficiency (t = −3.174, *p*-_Bonferroni_ = 0.008) (Figure 3D).

### 3.4. Brain Behavior Correlations

As shown in Figure 4, higher T-score of the inattention subscale in the Conners 3–PS were significantly correlated with lower nodal global efficiency in the left postcentral gyrus (r = −0.460, *p*-_Bonferroni_ = 0.015) and lower nodal degree in left inferior parietal lobule (r = −0.443, *p*-_Bonferroni_ = 0.020) in children with TBI-A at the early stimulation stage. No significant brain–behavior correlations were found in the group of controls.

## 4. Discussion

In this study, we found that relative to matched controls, the left precentral gyrus in children with TBI-A showed significantly lower capacity for functional information transferring (represented by significantly lower nodal clustering coefficient) during sustained attention processing. The precentral gyrus, as the location of the primary motor cortex, has been proven to be significantly involved in response inhibition function [67]. Previous fMRI studies in pediatric TBI have found significantly decreased precentral activation in children with TBI during performance of sustained attention [19], inhibitory control [20], and language processing [17]. Clinical studies have also suggested that children with TBI are vulnerable to deficits in inhibitory control [68,69], and children with post-TBI attention deficits have even more severe impairments than children with normal outcomes after TBI [70].

Meanwhile, the left parietal cortex, particularly the left postcentral and inferior parietal gyri, showed significantly suboptimal regional efficiency for functional communications with other brain regions during sustained attention processing, especially at the early stimulation stage, and these system-level functional anomalies associated with the left parietal cortex were found to greatly link to the severe inattentive symptoms in children with TBI-A. The parietal cortex is a key component of the attention network, involving in both the bottom-up selection and top-down control processes [71,72]. Within the attention network, the postcentral gyrus is responsible for transferring tactile information during the spatial attention [73], while the inferior parietal gyrus for information integration in the frontoparietal pathways during cognitive control [74]. Functional brain alterations associated with parietal cortex have been frequently reported in previous studies in children with TBI when performing tasks requiring attention [19,20], interference control [21], working memory [11,12], and motor control [16].

Indeed, both frontal and parietal lobes are core components subserving attention processing and cognitive control in the human brain. There has been growing consensus that dynamic disruptions of the frontal and parietal systems play the central role in chronical post-TBI neurocognitive and behavioral impairment, especially in the attention and cognitive control domains [75]. Together with the existing findings, results of the present study further suggest that the suboptimal efficiency of left parietal regions for functional interactions with other brain areas, especially during the initiation stage of attention processing, significantly implicate the TBI-A-specific impairment of the attention network, which can contribute to severe behavioral inattentiveness and hyperactivity/impulsivity in children with TBI.

Compared with the control group, our study also found that at the late-stimulation stage, the left fusiform gyrus in children with TBI-A had significantly decreased stability for maintaining the efficiency of functional interactions with other brain regions. The fusiform gyrus is part of the temporal and occipital lobes, which has been found to be a key structure for high-order visual (such as face, body, and high special frequency objects) and imagery processing [76,77]. Functional brain activation and connectivity studies in children with TBI have also reported fusiform gyrus-related abnormalities, such as reduced activations during sustained attention processing [22], significantly increased activation during working memory processing [11], and altered functional connectivity between the fusiform gyrus and the frontal lobe during resting state [25]. On the other hand, studies have demonstrated significant involvement of the fusiform gyrus in severe psychopathology, especially psychosocial and emotional dysregulation and thought problems in patients with major depressive disorder [78,79], schizophrenia [80,81], and other mental disorders [82,83]. In addition, there is growing evidence to support TBI as a risk factor for psychosis in both adult [84,85] and adolescent [86]. However, the underlying mechanisms are still fragmentary. On the basis of these prior studies from our and other groups, we hypothesize that post-TBI functional alterations associated with the left fusiform gyrus may significantly link to the development of severe late adolescence psychopathology, such as anxious/depressed, social and thought problems, in those with childhood TBI-A. Longitudinal follow-up of children with TBI-A will help to test this hypothesis.

There are several limitations associated with the current study. First, the sample size is relatively modest, which can limit the statistic power of the proposed analyses. Compared with other existing studies with similar sample sizes, the effect size of our study is relatively larger, because of the inclusion criteria of the two diagnostic groups (the T-scores of inattentive and hyperactive subscales were ≥65 for TBI-A, whereas ≤60 for controls). The increased effect size can help improve statistical power of our study. Future research with a larger sample size is expected to further validate the results. Second, due to the nature of the sliding window approach, functional dynamics analysis is highly sensitive to head motions. Therefore, we took extra precautions during the setup before each scan and applied a restrictive cut-off threshold of 1.5 mm to minimize potential errors caused by head motions. In addition, factors such as injury severity, number of injuries, and time interval between injury and study visit may introduce confounders of the results. Nevertheless, our supplementary analyses showed that the detected functional alterations did not show significant correlations with injury severity, number of injuries, and time intervals. Third, we applied the commonly used cluster-based thresholding method for network node region determination. This conventional thresholding method can mistakenly exclude regions with significantly activation, if their size were smaller than the cluster threshold. Therefore, future studies should consider a combination of the cluster size and activation density thresholds for network node region determination, with parameters being adaptively adjusted according to the size of the applied regions. Such improved thresholding method can better balance the potential false-positive and false-negative errors in the network node selection step.

## 5. Conclusions

In summary, the current study reported significant alterations of the topological properties of the sustained attention processing network and their temporal dynamics in children with severe post-TBI attention deficits, especially in the temporal and parietal regions. Additionally, these system-level functional alterations were significantly linked with the elevated inattentive behaviors in the group of TBI-A. These findings provide valuable insight into the neurobiological and neurophysiological substrates associated with the onset of post-TBI attention deficits in children. This study also provided positive evidence that analysis of functional network dynamics can demonstrate the temporal instability of the functional brain pathway characteristics of TBI-related attention deficits in children.

## Figures and Tables

**Figure 1 brainsci-11-01348-f001:**
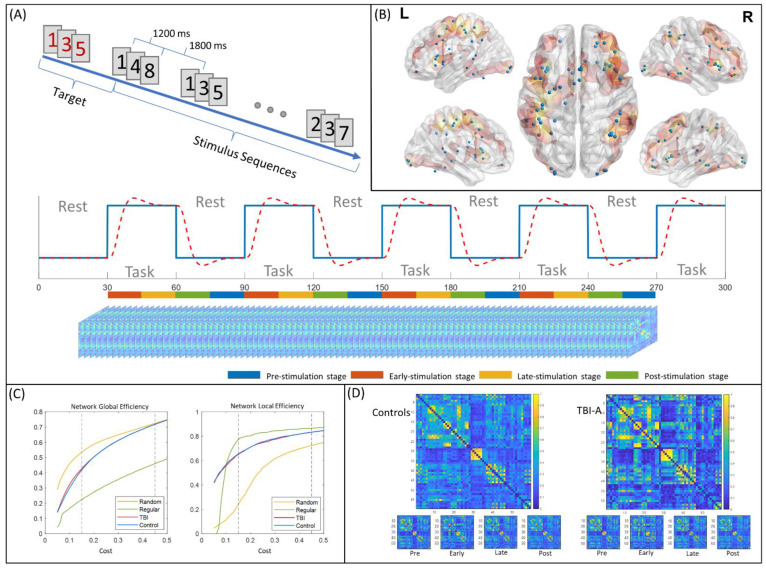
Functional network construction steps. (**A**) Block design and sub-stage definitions. (**B**) Selected nodes for functional network construction; (**C**) The network global and local efficiency curves of TBI-A and controls over the cost range of 0.05 to 0.5; (**D**) The group connectivity matrices of controls and TBI-A. L: left hemisphere; R: right hemisphere; TBI-A: children with severe post-traumatic brain injury attention deficits; Pre: pre-stimulation stage; Early: early stimulation stage; Late: late-stimulation stage; Post: post-stimulation stage.

**Figure 2 brainsci-11-01348-f002:**
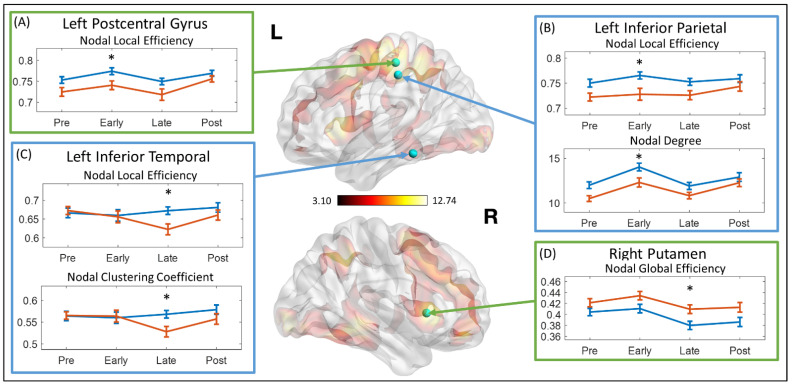
Mean network topological properties at different stages. The topological properties that showed significant group difference were marked with asterisk (*). (**A**) Mean nodal local efficiency at left postcentral gyrus. (**B**) Mean nodal local efficiency and mean nodal degree at left inferior parietal gyrus. (**C**) Mean nodal local efficiency and mean nodal clustering coefficient at left inferior temporal gyrus. (**D**) Mean nodal global efficiency at right putamen. L: left hemisphere; R: right hemisphere; Pre: pre-stimulation stage; Early: early stimulation stage; Late: late-stimulation stage; Post: post-stimulation stage.

**Figure 3 brainsci-11-01348-f003:**
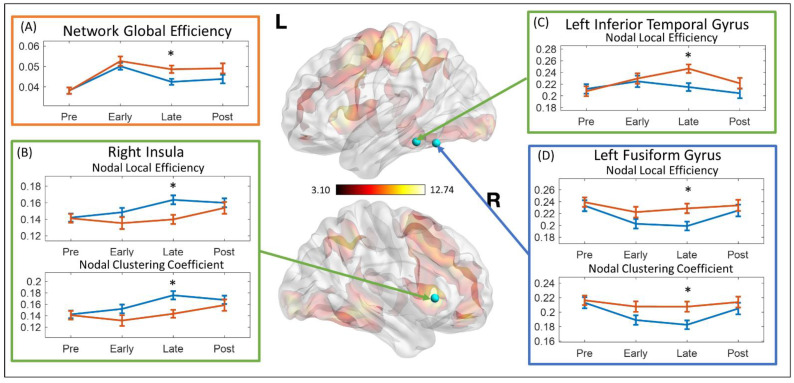
Standard deviation of network topological properties at different stages. The topological properties that showed significant group difference were marked with asterisk (*). (**A**) Standard deviation of the network global efficiency. (**B**) Standard deviation of the nodal local efficiency and the nodal clustering coefficient at right insula. (**C**) Standard deviation of the nodal local efficiency at left inferior temporal gyrus. (**D**) Standard deviation of the nodal local efficiency and the nodal clustering coefficient at left fusiform gyrus. L: left hemisphere; R: right hemisphere; Pre: pre-stimulation stage; Early: early stimulation stage; Late: late-stimulation stage; Post: post-stimulation stage.

**Figure 4 brainsci-11-01348-f004:**
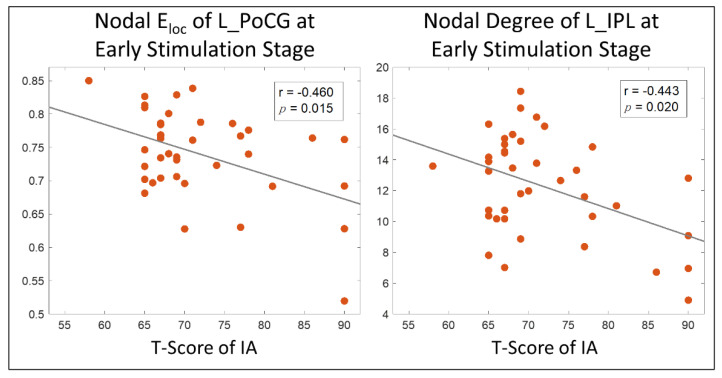
Brain-behavior correlation analysis results. E_loc_: local efficiency; L_PoCG: left postcentral gyrus; IA: inattention; L_IPL: left inferior parietal lobule.

**Table 1 brainsci-11-01348-t001:** Demographic and clinical characteristics of the study sample.

	Controls Mean (SD)	TBI-A Mean (SD)	t or *χ*^2^ Value	*p* Value
*n*	46	40		
Male/Female	26/20	24/16	0.106 (***χ*^2^**)	0.744
Socioeconomic Status ^1^	16.19 (1.80)	15.57 (4.19)	1.450	0.151
Full Scale IQ	115.00 (10.60)	110.48 (12.68)	1.802	0.075
Age	13.24 (1.47)	13.27 (1.72)	−0.075	0.940
Ethnicity/Race			2.411 (***χ*^2^**)	0.300
Caucasian	27	29		
Hispanic	8	3		
Others	11	8		
Conners 3–PS-based T Score				
Inattention	46.67 (6.22)	71.75 (8.09)	−16.366	<0.001
Hyperactivity/Impulsivity	47.91 (5.68)	63.50 (14.23)	−6.835	<0.001

^1^ Socioeconomic status was estimated using the average education year of both parents. TBI-A: children with traumatic brain injury related attention deficits; SD: standard deviation; n: number of subjects; Conners 3–PS: Conners 3rd Edition–Parent Short-Form T-score.

## Data Availability

Data are available upon reasonable request to the corresponding author.

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
