# Peer review of "Abnormal Functional Network Topology and Its Dynamics during Sustained Attention Processing Significantly Implicate Post-TBI Attention Deficits in Children"

_brainsci, 2021, doi:10.3390/brainsci11101348_

Round 1

Reviewer 1 Report

See file attached.

Reviewer 2 Report

In the present study the Authors aimed to utilize the graph theoretical technique (GTT)- and dynamic functional connectivity (FC)-based techniques to study the topological properties and their dynamics of the functional network for attention processing, and their relations with traumatic brain injury (TBI)-related attention deficits in a homogeneous group of children with TBI-A and matched controls.

Overall, I found this study timely, original, well conducted and scientifically sound. I have only some minor suggestions aimed to improve the high quality of the paper and these are outlined below:

1) A brief note on neurodevelopment and potential psychiatric consequences of TBI would be useful with appropriate reference (see doi 10.3390/brainsci11020275).

2) How many subjects were screened, but refused to participate? Please specify.

3) How prior or current ADHD was diagnosed? Only by parent reporting or with specific objective measures?

Author Response

Response to Reviewer 2 Comments

General: In the present study the Authors aimed to utilize the graph theoretical technique (GTT)- and dynamic functional connectivity (FC)-based techniques to study the topological properties and their dynamics of the functional network for attention processing, and their relations with traumatic brain injury (TBI)-related attention deficits in a homogeneous group of children with TBI-A and matched controls.

Overall, I found this study timely, original, well conducted and scientifically sound. I have only some minor suggestions aimed to improve the high quality of the paper and these are outlined below:

Response: We appreciate the very detailed and constructive comments from the reviewer. We have addressed all the comments and have made specific changes in the revised manuscript accordingly.

Point 1: A brief note on neurodevelopment and potential psychiatric consequences of TBI would be useful with appropriate reference (see doi 10.3390/brainsci11020275). 

Response 1: We greatly appreciate this valuable suggestion and have added the corresponding references in the revised manuscript in Introduction (page 2): “Neurocognitive impairments and behavioral abnormalities, including attention problems, depression and mood disorders, anxiety, and posttraumatic stress disorder, were frequently reported in children with chronic TBI [2-5]”, and Discussion (page 13): “In addition, there is growing evidence to support TBI as a risk factor for psychosis in both adult [84,85] and adolescent [86]. However, the underlying mechanisms are still fragmentary.

Reference

  1. Emery, C.A.; Barlow, K.M.; Brooks, B.L.; Max, J.E.; Villavicencio-Requis, A.; Gnanakumar, V.; Robertson, H.L.; Schneider, K.; Yeates, K.O. A Systematic Review of Psychiatric, Psychological, and Behavioural Outcomes following Mild Traumatic Brain Injury in Children and Adolescents. Can J Psychiatry 2016, 61, 259-269, doi:10.1177/0706743716643741.
  2. Hooper, S.R.; Alexander, J.; Moore, D.; Sasser, H.C.; Laurent, S.; King, J.; Bartel, S.; Callahan, B.J.N. Caregiver reports of common symptoms in children following a traumatic brain injury. 2004, 19, 175-189.3.
  3. Konigs, M.; Heij, H.A.; van der Sluijs, J.A.; Vermeulen, R.J.; Goslings, J.C.; Luitse, J.S.; Poll-The, B.T.; Beelen, A.; van der Wees, M.; Kemps, R.J.; et al. Pediatric Traumatic Brain Injury and Attention Deficit. Pediatrics 2015, 136, 534-541, doi:10.1542/peds.2015-0437.
  4. Polinder, S.; Haagsma, J.A.; van Klaveren, D.; Steyerberg, E.W.; van Beeck, E.F. Health-related quality of life after TBI: a systematic review of study design, instruments, measurement properties, and outcome. Popul Health Metr 2015, 13, 4, doi:10.1186/s12963-015-0037-1
  5. Fujii, D.E.; Ahmed, I. Psychotic disorder caused by traumatic brain injury. Psychiatr Clin North Am 2014, 37, 113-124, doi:10.1016/j.psc.2013.11.006.
  6. Molloy, C.; Conroy, R.M.; Cotter, D.R.; Cannon, M. Is traumatic brain injury a risk factor for schizophrenia? A meta-analysis of case-controlled population-based studies. Schizophr Bull 2011, 37, 1104-1110, doi:10.1093/schbul/sbr091
  7. Rabner, J.; Gottlieb, S.; Lazdowsky, L.; LeBel, A. Psychosis following traumatic brain injury and cannabis use in late adolescence. Am J Addict 2016, 25, 91-93, doi:10.1111/ajad.12338.

Point 2: How many subjects were screened, but refused to participate? Please specify.

Response 2: We initially screened the digital clinical databases of the New Jersey Pediatric Neuroscience Institute and Saint Peter’s University Hospital, advertised in local communities, and conducted phone screenings with about 150 parents. Among those potential candidates, over 1/3 had MRI constraints (teeth braces, etc) or other medical concerns that were exclusive in our study. According to our recruitment database, there were only a handful of parents raised concerns about MRI safety during our initial phone conversations, but later all agreed to participate in the study after being provided with written documents showing the safety of our study. There were no parents or children who were fully eligible but refused to participate in the study.

Point 3: How prior or current ADHD was diagnosed? Only by parent reporting or with specific objective measures?

Response 3: Prior ADHD was based on any medical report in the clinical database or by parent reporting. Current ADHD was based on based on DSM-5: having more than 4 inattentive or 4 hyperactive symptoms based on the newly validated KSADS-5.

Round 2

Reviewer 1 Report

The authors have addressed all my points.

I would just suggest the following minor points to their replies, which the authors may or may not include in the final version of the manuscript:

  • According to response 3, the authors have modified the text to include a detailed explanation of how a network is to be considered a small-world network. However, at no part in the manuscript do they mention that their aim was for networks to be of this kind. Thus, I would add something to the text to mention that this was the kind of property for the networks in the study that led their final choice of cost range.
  • Maybe a similar phrasing as response 6 could be added to the limitations paragraph in the Discussion section.
